# Research on the Mechanism of Corn Price Formation in China Based on the PLS-SEM Model

**DOI:** 10.3390/foods13060875

**Published:** 2024-03-14

**Authors:** Xin Shen, Cancheng Qiu

**Affiliations:** School of Economics and Management, Shanghai Ocean University, Shanghai 201306, China; qcc245664495@163.com

**Keywords:** corn prices, price formation mechanism, price transmission, structural equation modeling

## Abstract

After the cancellation of the temporary corn storage system in 2016, the price of corn in China returned to market regulation, resulting in increased price volatility. This study focuses on monthly data from April 2016 to March 2023 in China. The partial least squares structural equation modeling (PLS-SEM) is employed to analyze the impact of corn supply and demand factors, substitute prices, monetary supply, international corn prices, and international energy prices on the volatility of corn prices in China. Results indicate that supply and demand factors are the most direct influencers of corn prices, with demand factors having the most significant impact. Monetary supply and substitute prices affect corn prices through the demand side. External factors mainly consist of international energy and corn prices. The impact of international energy on Chinese corn prices is achieved through international corn prices, which directly influence the prices in China. It is recommended to stabilize corn market prices by regulating corn supply and demand, to improve the monitoring and early warning mechanisms for international energy and corn prices, and to implement measures for prudent regulation of monetary supply.

## 1. Introduction

According to BricBigData, China’s corn production reached 200 million tons in 2012, accounting for over one-third of the total grain yield, making it the most important crop in China. As of 2017, corn feed consumption in China reached 140 million tons, while industrial consumption reached 80 million tons, highlighting the pivotal role of corn in China’s agricultural sector. Furthermore, China holds a significant position and influence in the global corn industry. According to the statistical database of the Food and Agriculture Organization of the United Nations, China accounted for 25.38% of the global corn production in 2020, making it the second-largest corn producer globally, only after the United States. With the increasing interconnectedness of international trade, China’s economic ties with other countries around the world continue to strengthen, amplifying the impact of price fluctuations in the Chinese corn market on the global food market. The report, “Analysis of China’s Corn Industry Data in 2023” points out that China’s corn imports account for over 10% of global imports, and fluctuations in Chinese corn prices have a significant impact on international corn prices and other grain prices. Therefore, the stability of corn prices in China holds great significance for the global food market.

However, the stability of corn prices is facing challenges due to the reform of China’s temporary corn storage system. In 2008, China implemented a temporary corn storage system in the three northeastern provinces and the Inner Mongolia Autonomous Region. The government purchased a portion of corn at planned prices as part of this policy. The primary objectives of this policy were to achieve increased grain production, higher income for farmers, and to stabilize corn prices by regulating the supply in the market. While the implementation of this policy led to increased benefits for Chinese farmers and maintained stable corn prices, it also had some adverse effects. On the one hand, there was an issue of excessive corn stockpiling, resulting in a significant rise in corn storage costs and a substantial increase in fiscal pressure. On the other hand, the protected status of corn prices in China led to them remaining high, far above international corn prices, which caused domestic corn to lack competitiveness in the international market. Due to the issues at hand, in 2016, the National Development and Reform Commission of China discontinued the temporary corn storage system and introduced a new policy of “market-oriented procurement with subsidies”. Under this new policy, the government purchases surplus corn from farmers at real-time market prices and provides subsidies to corn producers, replacing the previous practice of purchasing corn at planned prices higher than the market price. With the implementation of this new policy, the price of corn is now determined by market forces and subject to market regulation.

Corn, characterized by its wide-ranging applications and high degree of financialization, is highly susceptible to various external factors and shocks in terms of price fluctuations after returning to market regulation. The dramatic fluctuation of corn prices during the period of 2015–2016 has sparked extensive academic discussions and garnered notable attention from the government. With the implementation of the “market-oriented purchasing with subsidies” policy in China in 2016, corn prices began to experience frequent oscillations. The intensified fluctuations in corn prices were further exacerbated by the impact of the 2020 pandemic and shifts in pig farming policies. According to the data from China’s Ministry of Agriculture, the average market price of corn soared by 17%, from CNY 2.319 per kilogram in 2020 to CNY 2.713 per kilogram in 2023. This significant price surge was largely attributed to the global rise in agricultural commodity prices caused by the Ukrainian crisis in 2022, as Ukraine is a major source of imported corn for China.

The fluctuation of corn prices, as a staple agricultural commodity, has become an issue of urgent concern for policymakers. The sharp rise in food prices over the past decade has pushed millions of people into poverty [1]. The dramatic oscillation of corn prices not only undermines the household income of corn farmers but also impacts the downstream industries of corn processing, adversely affecting the sound operation of the macroeconomy. The instability in corn market prices may also impede agricultural investment, reduce agricultural productivity, and have detrimental effects on impoverished farmers.

In order to explore and address fluctuations in the price of corn to ensure price stability, this paper aims to investigate the factors and mechanisms influencing maize prices. Given the multifactorial nature of maize price formation and the complexity of the relationships among various influencing factors, this study adopts a structural equation modeling approach capable of handling complex causal relationships between multiple variables [2]. By combining empirical research findings with real-world circumstances, this paper presents relevant recommendations.

## 2. Literature Review

The significant fluctuations in corn prices in China have emerged as a result of agricultural policy reforms, market liberalization, and economic globalization. As an essential agricultural commodity, the price of corn is intricately linked to the stability of other agricultural product prices, serving as an “anchor” for ensuring price stability within the agricultural sector. Consequently, stabilizing corn prices holds vital significance in maintaining stability in the prices of agricultural products within a nation.

Currently, research on corn prices can be broadly categorized into international and domestic influencing factors. Numerous scholars have investigated various factors affecting the price of corn from these two dimensions, utilizing methods such as VAR [3], ARCH [4], linear Granger causality [5], DCCA [6], and BEKK [7].

At the international level, the main factors that impact domestic corn prices are international corn prices and international energy prices. With the deepening of global trade, barriers between domestic and international markets are gradually being dismantled, leading to increased transmission effects among international agricultural products. Arnade et al. [8] explored the transmission relationship between US agricultural prices and Chinese agricultural prices, highlighting that agricultural products with lower import barriers are more susceptible to international price transmission. Ceballos et al. [9] conducted a study on the monthly price volatility of domestic and international corn markets in 27 countries using the GARCH method and observed a significant impact of the international corn market on the domestic corn market.

The impact of international energy on domestic corn prices differs from that of international corn prices and is mainly transmitted through increased production costs and the substitution of biomass energy. Rising energy prices lead to higher input costs and transportation costs for agricultural production, ultimately resulting in an increase in corn prices [10]. Since corn is a primary raw material for biomaterial energy, an increase in biomaterial energy demand may lead to a rise in corn prices. Some scholars have validated the entire transmission path from the energy market to the biomass fuel market and then to the corn market [11]. Others have used co-integration analysis and causal analysis methods to explore the long-term and short-term relations between crude oil, biomass fuel, and agricultural product prices, discovering significant short-term bi-directional causal relationships between WTI crude oil, corn ethanol, and corn prices [12].

Research on the domestic level primarily focuses on analyzing the role of government policies. Cummings et al. [13] indicate that food prices are influenced by public policies, citing that the price stability policy implemented by the Indian government keeps agricultural product prices stable within a certain range, helping farmers benefit from food cultivation through price policies. Monetary policy is another important factor contributing to changes in agricultural product prices. Monetary policy can lead to overshooting effects in agricultural products, where changes in the currency supply result in larger price increases compared to industrial goods, ultimately causing a significant overall increase in agricultural product prices. Asfaha and Jooste [14] pointed out the significant impact of monetary policy on agricultural commodity prices. A study on South African residents found that the adjustment speed of agricultural prices is faster than that of industrial prices when there is a change in circulating currency. Amatov and Dorfman’s [15] research indicated a significant positive correlation between the Federal Reserve's monetary policy and agricultural price indices. Research on China’s corn policy primarily focuses on the policies related to corn. Under the temporary purchase and storage system, the market regulation mechanism for corn is distorted by government policy prices. Even if there is a significant increase in corn supply, the government will still purchase corn at prices higher than the market price. As a result, the actual corn market prices are not accurately reflected by the corn prices [16].

The price of corn is influenced by factors such as livestock product prices, production costs, substitute prices, and the natural environment. Being a primary source of animal feed, corn demand may decrease when there is an oversupply or decline in demand for livestock and agricultural products, which can lead to a decrease in corn prices. Through the application of asymmetric BEKK-MGARCH and DCC-MGARCH models, Pan et al. [17] investigated the mechanisms of price fluctuations and dynamic correlations between the corn and pork market. They discovered bidirectional price spillover effects between the corn market and the pork market. The fluctuations in fertilizer prices primarily affect the price of corn through variations in production costs, although this impact is typically short-term [18]. Esposti and Listorti [19] examined the relationship between corn and substitute prices and found that corn prices have a positive response to changes in durum wheat prices, suggesting substitutability between corn and durum wheat. Muflikh et al. [20] explored chili pepper prices in Indonesia and identified factors such as weather conditions, water resource constraints, and pest outbreaks that could lead to significant fluctuations in chili pepper prices.

The complexity of corn prices is further compounded by the intricate interplay between various influencing factors. As noted by Sun [21], with the acceleration of China’s internationalization process, the factors influencing corn prices have become increasingly complex, with interdependencies among them. For instance, international energy prices not only influence international corn prices but also impact China’s corn prices through cost effects. Moreover, international corn prices can be transmitted to domestic corn prices through international trade channels.

Existing literature has extensively examined the various factors that impact the formation of agricultural product prices, which is crucial for the establishment and improvement of agricultural price systems. However, relatively few studies have focused on the determinants of corn prices, with most of the existing research concentrating on the transmission of crude oil prices and domestic and foreign corn prices, with an emphasis on analyzing individual influencing factors. Xiarchos and Burnett [22] adopted the spillover index methodology to examine the relationship between energy futures and corn prices. By controlling for market trends and seasonal factors, they identified a significant impact of energy futures prices on corn prices, thereby confirming the influential role of energy futures prices. Kocak et al. [23] utilized a non-linear smoothing transition model to investigate the effects of ethanol, renewable biofuels, oil prices, population, and exchange rates on the price of corn in the United States. Although these studies examined the impacts of various factors on corn prices, they did not delve into the complex relationships among these influencing factors, leaving the mechanisms behind corn price formation unclear. Hence, it is necessary to conduct in-depth research on both the influencing factors and the mechanisms underlying the formation of corn prices.

China’s corn market holds a significant position in the international corn market. Hence, this study aims to investigate the formation mechanism of corn prices in the Chinese market. Taking into account the impact of the temporary storage system implemented by China between 2008 and 2015 on corn prices, this study will utilize data from 2016 to 2023 to conduct research. By exploring the factors and mechanisms influencing corn prices, this paper aims to provide references and guidance for stabilizing corn prices.

## 3. Research Hypothesis

To explore and respond to corn price fluctuations while ensuring price stability, this article uses a structural equation model to analyze the influencing factors and formation mechanisms of corn prices and provides corresponding suggestions based on empirical research results and real-world conditions. The price of corn is influenced by a multitude of factors. This paper explores the impact of six factors on corn prices, including corn demand, corn supply, substitute prices, currency supply, international corn prices, and global energy prices.

### 3.1. Corn Demand

According to data from BricBigData, in 2021, feed consumption and industrial consumption accounted for 66% and 26% of total corn consumption in China, respectively, with industrial consumption being the main source of corn demand. An increase in corn demand will drive up corn prices. Based on the economic principle of supply and demand, when the supply remains unchanged, an increase in demand will lead to a sustained increase in corn prices.

**Hypothesis 1 (H1).** 
*There is a positive impact of corn demand on corn prices.*


### 3.2. Corn Supply

Corn supply is typically composed of production, beginning stocks, and imports, with production being the dominant factor. According to BricBigData, corn production accounted for 64.7% of the total corn supply in 2021. Therefore, total corn production is used as the measurement indicator for corn supply.

Under the strong protection of the temporary stockpiling policy, corn cultivation remains profitable even with extensive planting, leading to a continuous increase in cultivation area. During this period, market regulation on corn is relatively limited. Following the reform of the temporary stockpiling system, corn prices are primarily influenced by supply and demand dynamics. When demand remains constant, an increase in corn supply would result in a decrease in corn prices.

**Hypothesis 2 (H2).** 
*Corn supply has a negative impact on corn prices.*


### 3.3. Substitute Price

Corn substitutes refer to products that are used in place of corn for edible and feed processing. Wheat is the main grain, while soybean and soybean meal are the main raw materials for feed processing. Therefore, wheat, soybean meal, and soybean are selected as the price indicators for corn substitutes. The price increase of alternative products has led to corresponding changes in the prices of both premium agricultural products and agricultural by-products within the same supply chain [24]. When prices of corn substitutes rise, consumers tend to purchase more corn for food consumption, while businesses engaged in feed processing also rely more on corn as a primary ingredient—consequently, the overall demand for corn increases.

**Hypothesis 3 (H3).** 
*Substitute prices have a positive impact on corn demand.*


### 3.4. Currency Supply

The earliest scholars to explore the impact of monetary policy on agricultural product prices from a theoretical perspective were Bordo [25] and Frankel [26]. They proposed the “Contract Length Hypothesis” and the “ Overshooting Hypothesis of Agricultural Prices” respectively. Both hypotheses demonstrate that monetary shocks lead to larger changes in agricultural product prices compared to industrial product prices, providing evidence for the significant influence of monetary policy on agricultural product prices from different angles. Among them, the overshooting hypothesis offers a more comprehensive explanation as it further suggests that agricultural product prices will overshoot their new equilibrium values in the short term. This hypothesis highlights that monetary policy shocks, through nominal interest rates, result in speculative changes in the demand for agricultural products, leading to rapid adjustments in agricultural product prices while industrial product prices adjust more slowly. Hence, it indicates that monetary supply is non-neutral in the short term.

When the currency supply increases, market transactions become more active, and the increase in available liquid funds stimulates consumption, thereby increasing the demand for agricultural products. With the development of agricultural futures markets and the increasing financialization of agricultural products, the impact of currency supply has also deepened. Highly financialized agricultural commodities such as soybeans and wheat are highly vulnerable to changes in the money supply, and loose monetary policy can trigger large speculative demand, leading to rapid price increases in these agricultural commodities.

**Hypothesis 4a (H4a).** 
*There is a positive relationship between currency supply and corn demand.*


**Hypothesis 4b (H4b).** 
*There is a positive relationship between currency supply and substitute price.*


### 3.5. International Corn Price

Price transmission among different markets plays a crucial role in assessing market efficiency [8]. With the deepening of international trade relations, integration between domestic and foreign agricultural markets has been increasingly evident, resulting in a heightened degree of overall price transmission.

Since its accession to the World Trade Organization, China has embraced a relatively liberal trade system, leading to a growing connection between domestic and international maize prices and a convergence of price trends. The international corn market exerts its influence on domestic corn consumption prices primarily through the mechanism of grain imports. When imported corn is used as feed or for industrial purposes, international corn prices will affect domestic processed goods by influencing the production costs of industrial products. This, in turn, leads to a comprehensive increase in food prices.

International corn not only affects the domestic price of corn but also exerts an influence on the prices of other major crops, such as soybeans and wheat, through international trade, as these agricultural products exhibit strong substitutability with corn.

The international price of corn also has an impact on the domestic supply of corn. When the international price of corn rises, the high-priced imported corn will impact the domestic corn market, leading to an increase in the demand for domestic corn for consumption and processing. This increased demand will, in turn, prompt farmers to increase their cultivation of corn.

**Hypothesis 5a (H5a).** 
*International corn prices have a positive impact on domestic corn prices.*


**Hypothesis 5b (H5b).** 
*International corn prices have a positive impact on the substitute price.*


**Hypothesis 5c (H5c).** 
*International corn prices have a positive impact on corn supply.*


### 3.6. International Energy Price

International energy indirectly influences domestic corn prices through global corn prices. The transmission relationship between energy prices and corn prices is a long-standing and asymmetrical one, where the impact of rising energy prices on corn prices is greater than that of falling energy prices [27].

The influence of international energy on international corn prices is realized through two pathways: production costs and substitution with biomass energy [28]. Energy plays a crucial role in corn production, serving as the primary raw material for fertilizers and powering the production and transportation of corn. Rising energy prices lead to increased production costs for corn, thereby driving upward pressure on corn prices. Moreover, corn can be utilized as a raw material for biomass energy production and exhibits a substitutive relationship with petroleum, resulting in a positive correlation between their respective prices.

**Hypothesis 6 (H6).** 
*International energy prices have a positive impact on international corn prices.*


Taking into account the aforementioned hypotheses, this study considers monetary supply and international energy prices as exogenous factors influencing corn prices, while corn price, corn supply, corn demand, substitute prices, and international corn prices are regarded as endogenous factors. The proposed model for corn price formation is illustrated in Figure 1.

## 4. Model Establishment and Data Sources

### 4.1. Model Establishment and Explanation

Currently, research on prices primarily utilizes the VAR method to study price fluctuations, with a greater emphasis on analyzing the impact of individual factors on prices. However, in order to provide a more comprehensive reflection of the current economic situation, this paper examines the relationship between corn prices and multiple variables using structural equation modeling.

Originally applied in sociology and management research, SEM has gradually been applied in the field of economics, expanding the methodological framework of economic research. SEM is particularly suitable for testing complex hypotheses of interdependencies among multiple variables since it integrates factor analysis and path analysis, serving as a multivariate statistical analysis method [29]. Given the complex interrelationships among the factors influencing corn prices, SEM aligns well with the conditions required for its application.

In this study, the partial least squares structural equation modeling (PLS-SEM) method is employed to validate the mechanism of corn price formation.

Contrasting with covariance-based structural equation modeling (CB-SEM), partial least squares structural equation modeling (PLS-SEM) requires a smaller sample size [30,31] and can handle non-normal data [32]. Additionally, PLS-SEM excels at addressing issues of multicollinearity, making it highly suitable for exploratory analysis and validation of constructed model structures. It offers advantages in exploratory and predictive analyses within the realm of complex research.

Due to the relatively small sample size (84) and the non-normal distribution of the data in this study, as well as the need to explore complex structural relationships, the PLS-SEM model has been chosen. The SmartPLS 4.0 software will be utilized for data analysis and processing.

### 4.2. Variable Description and Data Source

This study utilized 84 monthly data points from April 2016 to March 2023. The analysis commenced in April 2016, as it marked the discontinuation of the temporary corn procurement system in China. Subsequently, corn prices began to be regulated by market forces, allowing for a study of corn prices without the interference of procurement policies. Daily price data was transformed into monthly price data by calculating the average values for each month. Missing data points were supplemented through interpolation methods. To mitigate the impact of level and scale differences between variables, the data was standardized prior to analysis. Please refer to Table 1 for further details regarding the specific selection of latent and measured variables, as well as the sources of data.

The corn price represents the market price of corn in China, and the data for measuring it is sourced from the National Bureau of Statistics of China.

Corn supply represents the overall supply of corn in China and is mainly composed of production, stocks, and imports. Considering that corn production constitutes a major component of China’s corn supply, this study selects total corn production as the measurement variable for corn supply. The data for corn production is obtained from the monthly data published by the National Bureau of Statistics in China.

Corn demand refers to the total demand for corn in the country, measured by the total consumption of corn and the price of corn starch. The total consumption of corn represents the domestic consumer demand for corn, while the fluctuation in the price of corn starch corresponds to changes in the demand for corn industrial processing. The monthly data for these two variables are sourced from the National Bureau of Statistics of China.

Substitute prices refer to the prices of agricultural products that can be used as substitutes for corn. In this study, we have selected soybeans, wheat, and soybean meal as the measuring variables, as they exhibit strong substitutability with corn. The national average spot price was selected for soybean and wheat prices, and the market price of soybean meal was selected for soybean meal prices. The monthly data for these three variables are sourced from the National Bureau of Statistics of China.

International corn price refers to the price level of corn in the international market. The Chicago Board of Trade (CBOT), as a major futures market, is a mainstream exchange for corn futures trading that reflects real-time prices in the international corn market. Therefore, this study selects the CBOT corn price data published by the Chicago Board of Trade as the measure.

International energy prices represent the prices of major energy sources in the international market. The selected measuring variables mainly include natural gas prices, crude oil prices, and biomass energy prices, which are representative energy sources. Natural gas prices are taken from the New York Commodity Exchange (NYMEX) natural gas futures prices. Crude oil prices include West Texas Intermediate (WTI) crude oil prices and Brent crude oil futures prices, and data for these three energy sources are sourced from NYMEX. The biomass energy price selected is for blended biodiesel, and the data is sourced from the Chicago Board of Trade.

International energy prices refer to the prices of major energy sources in the international market. The selected variables include representative prices of natural gas, crude oil, and biomass energy. For natural gas prices, data on NYMEX natural gas futures prices are used. For crude oil prices, the data of West Texas Intermediate (WTI) crude oil prices and Brent crude oil futures prices are selected, and the data source for these three energy prices is the New York Mercantile Exchange. The price of biomass energy is based on the price data of blended biodiesel, which is from the official website of the Chicago Board of Trade.

Currency supply refers to the total amount of currency issued by a country or region within a certain period of time. The measurement of currency supply typically focuses on M0 and M1, which represent different forms of circulating currency. M0 represents physical cash in circulation, while M1 encompasses both cash and demand deposits that are readily available for transactions. Data pertaining to currency supply is sourced from the National Bureau of Statistics of China.

## 5. Empirical Test and Result Analysis

### 5.1. Measurement Model Test

In the analysis of structural equation modeling (SEM), this study firstly examines the reliability and validity of the measurement model and then tests the explanatory power of the structural model and the significance of path coefficients. Reliability and validity tests can assess the internal consistency and convergence of the measurement model, which are the main methods for evaluating the effectiveness of the measurement model.

Cronbach’s α is a commonly used method to assess reliability, calculated as α = (k/(k − 1)) × (1 − (∑Si^2^)/ST^2^). It reflects the consistency among the observed indicators, and a Cronbach’s α value greater than 0.7 for each latent variable indicates an acceptable range of reliability [30]. The composite reliability (C.R.) test generally requires a C.R. value greater than 0.7 for acceptance, and the average variance extracted (AVE) should exceed 0.5 [33].

Table 2 displays the results of the measurement model assessment. The range of Cronbach’s α coefficients is 0.821 to 0.945, all exceeding the threshold of 0.7. The range of composite reliability (C.R.) values is 0.822 to 0.954, all surpassing the standard criterion of 0.6. Furthermore, the average variance extracted (AVE) values all exceed the threshold of 0.50. These results indicate that the measurement model has passed the reliability test, demonstrating that the selected indicators effectively reflect their corresponding latent constructs.

In structural equation modeling, validity refers to the accuracy and effectiveness of measurement. A construct demonstrates discriminant validity when the square root of the average variance extracted (AVE) for each latent variable is greater than the square root of the correlations between that construct and other constructs [34]. In Table 3, the absolute values on the diagonal are greater than the corresponding absolute values in the rows and columns, indicating good discriminant validity of the measurement model. Therefore, the measurement model has passed the tests of reliability and validity, enabling further analysis.

### 5.2. Structural Model Test

#### 5.2.1. Test of Explanatory Power of the Model

In structural equation modeling, the explanatory power of a model is primarily assessed using R-squared (R^2^) and Q-squared (Q^2^). R^2^ measures the extent to which endogenous latent variables are explained, with values between 0.19–0.33, 0.33–0.67, and above 0.67, indicating weak, moderate, and strong explanatory power, respectively [34]. Q^2^ represents the predictive relevance of the model, with Hair et al. stating that Q^2^ values greater than 0 indicate high model relevance [30]. In Table 4,in this study, the Q^2^ values for all latent variables are greater than 0, indicating a strong predictive relevance of the exogenous variables on the endogenous variables.

The R² for the supply of corn is greater than 0.19, indicating a relatively low level of explained variation in the endogenous latent variable. The R^2^ values for corn demand, substitute prices, international corn prices, and corn prices are all greater than 0.67, with corn prices having an R^2^ of 0.898. This suggests that these endogenous latent variables are well-explained. The R^2^ values for all endogenous latent variables are greater than 0.19, indicating strong explanatory power of the model.

#### 5.2.2. Test of Path Coefficient

The data analysis was conducted using SmartPLS 4 and SPSS Statistics 26. After adjusting and screening the indicators, the final results are presented in Figure 2. In the structural equation model, all path coefficients and factor loadings meet the requirements of the model.

The Bootstrap method, using partial least squares (PLS), was employed to perform repeated sampling with replacement on the original data. The significance of individual path coefficients was tested using the T-statistic. Table 5 presents the path relationships of corn prices, with all path coefficients having T-values greater than 1.96. These results are considered significant, providing evidence to support the acceptance of hypotheses H1–H6.

From a supply and demand perspective, it can be analyzed that demand has a decisive impact on determining the price of corn, with a path coefficient of 0.837. Conversely, an increase in corn supply has a negative effect on its price, with a path coefficient of −0.256, meaning that for every unit increase in corn supply, there is a corresponding decrease of 0.256 units in corn price. Following the reform of the corn storage system, the price of corn will obey the law of supply and demand, with an increase in corn demand resulting in a sustained upward surge in corn prices, while an increase in supply will cause a decrease in corn prices.

From a domestic impact perspective, the currency supply and substitute prices primarily affect corn prices through corn demand. With respect to the currency supply, the factor loading of M0 is greater than that of M1. M0 refers to the total amount of cash in circulation in a country or region’s current market. This result suggests that extensive cash circulation has a greater impact on corn demand.

At a 1% significance level, the path coefficients of the currency supply and substitute prices on corn demand are 0.758 and 0.181, respectively. Both an increase in the currency supply of liquid assets and a rise in substitute prices will ultimately increase corn consumption and processing demands. These two factors serve as essential indirect factors that impact corn prices.

Furthermore, the currency supply also affects substitute prices, with a path coefficient of 0.175. Soybeans, wheat, and bean meal are highly financialized agricultural products, especially soybeans, which are among the most active agricultural futures contracts. When the currency supply increases, it can drive up the prices of these agricultural products.

From an international impact perspective, research has found that there are multiple transmission paths from international corn prices to domestic corn prices. In addition to the direct impact, international corn prices can also affect domestic corn prices through corn supply and substitute prices. The path coefficients of international corn prices on domestic corn prices, corn supply, and substitute prices are 0.327, 0.53, and 0.808, respectively. International corn prices can be transmitted to domestic corn prices through market mechanisms. At a 5% significance level, a 1% increase in international corn prices leads to a 0.327% increase in domestic corn prices.

The impact of international corn prices on corn supply is significant. With the rise in international corn prices, the quantity of imported corn decreases correspondingly, and high-priced international corn becomes less competitive domestically. This increases the enthusiasm of domestic farmers to cultivate corn, leading to an increase in corn supply. It is worth noting that the path coefficient of international corn prices on substitute prices is the highest. This is primarily because soybeans are the most financially integrated agricultural product globally, and the impact of international corn on domestic agricultural products is mainly through futures trading markets.

The path coefficient of international energy on international corn prices is 0.891, indicating a significant impact. International energy affects international corn prices primarily through cost and biomass energy transmission. When international energy prices increase, it will lead to an increase in corn production costs. Additionally, a rise in biomass energy prices will greatly increase the demand for corn raw materials, leading to an increase in corn prices.

#### 5.2.3. Mediating Effect Test

To test the mediating effects present in the model, this paper utilized the Bootstrapping resampling method. Using both the B-C method and percentile-based methods for significance testing at the 95% confidence interval, it was found that the upper and lower limits of the confidence interval did not include 0. The results are presented in Table 6. Thus, it can be concluded that the mediating effects of corn supply, corn demand, international corn prices, and substitute prices are significant.

## 6. Conclusions and Implications

The price fluctuations of corn, as an important food crop, can have a significant impact on the agricultural income of farmers, the development of the livestock industry, and the growth of agricultural processing enterprises. It can also pose a certain threat to China’s food security. Therefore, it is necessary to regulate the price of corn in China to ensure its relative stability.

This research is based on the data collected after the reform of China’s temporary corn storage system. By employing the partial least squares-structural equation model (PLS-SEM), the study investigates the price dynamics of corn in China under market regulation and explores the mechanisms behind its formation. The main findings are as follows:

Firstly, the overall fluctuation of corn prices in China is profoundly influenced by several factors, namely corn supply, corn demand, monetary supply, substitute prices, international energy prices, and international corn prices. However, it is important to note that these factors vary significantly in terms of their impact intensity on the trajectory of grain prices in our country.

Corn demand serves as the primary driver behind the volatility of corn prices, exerting the greatest influence on their fluctuations. Conversely, supply factors have a relatively smaller impact on corn prices. Supply and demand are the most direct factors affecting price; thus, adjusting the balance between supply and demand becomes crucial in price regulation. According to data from BricBigData, the difference between corn supply and demand in China remained between 30 million and 60 million metric tons from 2000 to 2010, contributing to the overall stability of corn prices during this period. Therefore, maintaining a state where supply slightly exceeds demand is beneficial for price stability. Controlling the difference between supply and demand within this range is recommended for future corn production.

The government can provide fiscal subsidies or other incentive measures to stimulate the growth of corn consumption in the food, feed, and processing sectors, thereby expanding the overall demand for corn in the market. When the supply of corn is low, the government can enhance the production capacity in the main corn-producing regions and optimize the corn cultivation structure to increase the supply of corn. Control over corn supply can be achieved by increasing production capacity in the main corn-producing areas and optimizing the corn planting structure. These measures can help maintain a slight surplus in corn supply, further balancing the supply and demand relationship and reducing price fluctuations.

Secondly, there is a significant interconnection among various factors influencing corn prices. By employing structural equation modeling (SEM), with its unique structural model construction and path relationship analysis, this study investigates the relationships among these influencing factors from a systematic perspective. These interrelationships between factors render the mechanism of corn price influences more complex, thus necessitating governmental departments to consider these interconnections when stabilizing corn prices and ensuring food security.

International corn prices can indeed impact domestic corn supply and substitute product prices. Therefore, while monitoring international corn prices, the government should not only consider the transmission path of international corn prices to domestic corn prices but also take into account the impact of international corn prices on other agricultural product prices, especially the impact on highly financialized soybean and wheat prices. Furthermore, international energy prices can indirectly affect domestic corn prices through their impact on international corn prices. On the one hand, an increase in energy prices leads to an increase in corn prices through cost-driven effects. On the other hand, an increase in biomass energy prices indirectly raises corn prices through demand-driven effects. Hence, policymakers need to be aware of this chain transmission mechanism between international energy and international corn prices and properly manage the spillover fluctuations in international energy markets to maintain long-term stability in the corn market.

The supply of currency can have an impact on the prices of corn substitutes and the demand for corn. An increase in the money supply can stimulate higher income levels, leading to increased demand for corn. At the same time, it can also result in rising costs of agricultural production inputs, driving up the prices of corn substitutes. Therefore, policymakers should fully consider the various effects of money supply on corn prices when using monetary policy to regulate and control.

## Figures and Tables

**Figure 1 foods-13-00875-f001:**
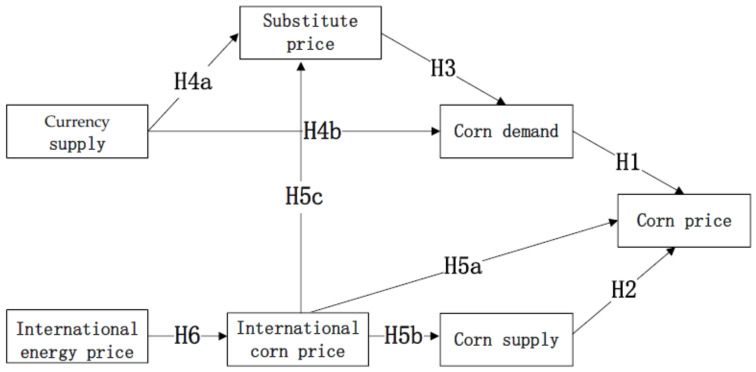
Structural equation model of corn prices.

**Figure 2 foods-13-00875-f002:**
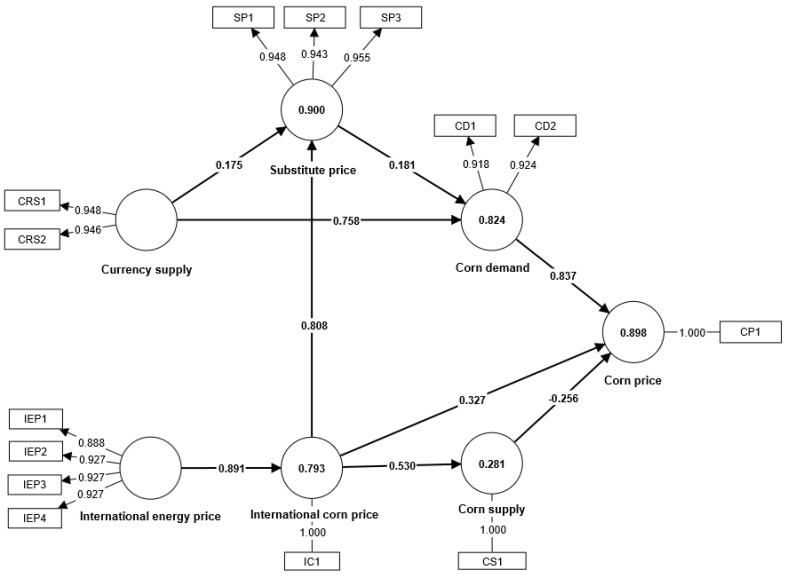
Results of the structural equation model operation.

**Table 1 foods-13-00875-t001:** Variable description and data sources.

Latent Variable	Measured Variable	Variable Code	Data Source
Corn price	Corn market price	CP1	National Bureau of Statistics of China
Corn supply	Total corn production	CS1	National Bureau of Statistics of China
Corn demand	Total corn consumption	CD1	National Bureau of Statistics of China
	Corn starch price	CD2	
International energy price	International gas price	IEP1	New York Commodity Exchange; Chicago Board of Trade
	WTI crude oil futures price	IEP2	
	Brent crude oil futures prices	IEP3	
	Biomass energy price	IEP4	
International corn price	CBOT corn futures price	IAP1	Chicago Board of Trade
Currency supply	M0	CRS1	People’s Bank of China
	M1	CRS2	
Substitute price	Wheat price	SP1	National Bureau of Statistics of China
	Soybean price	SP2	
	Soybean meal price	SP3	

**Table 2 foods-13-00875-t002:** Reliability and convergent validity analysis.

Latent Variable	Cronbach’s α	C.R.
International energy price	0.938	0.949
Substitute price	0.945	0.954
Corn demand	0.821	0.822
Currency supply	0.886	0.886

**Table 3 foods-13-00875-t003:** Construct validity analysis.

	International Corn Price	International Energy Price	Substitute Price	Corn Price	Corn Supply	Corn Demand	Currency Supply
International corn price	1						
International energy price	0.891	0.918					
Substitute price	0.942	0.899	0.949				
Corn price	0.863	0.724	0.835	1			
Corn supply	0.53	0.53	0.528	0.552	1		
Corn demand	0.803	0.732	0.781	0.905	0.758	0.921	
Currency supply	0.764	0.711	0.792	0.848	0.662	0.901	0.947

**Table 4 foods-13-00875-t004:** Goodness of fit and predictive correlation test.

Latent Variable	R^2^	Q^2^
International corn price	0.793	0.783
Substitute price	0.900	0.794
Corn price	0.898	0.885
Corn supply	0.281	0.277
Corn demand	0.824	0.690

**Table 5 foods-13-00875-t005:** Path coefficient test.

Relation of Path	Hypothesis	Coefficient of Path	Value of T
Corn demand -> Corn price	H1	0.837	10.39 ***
Corn supply -> Corn price	H2	−0.256	5.13 ***
Substitute price -> Corn demand	H3	0.181	3.541 ***
Currency supply -> Substitute price	H4a	0.175	4.228 ***
Currency supply -> Corn demand	H4b	0.758	16.576 ***
International corn price -> Corn price	H5a	0.327	5.121 ***
International corn price -> Corn supply	H5b	0.53	6.812 ***
International corn price -> Substitute price	H5c	0.808	20.351 ***
International energy price -> International corn price	H6	0.891	37.169 ***

Note: *** indicate significance at the levels of 0.001.

**Table 6 foods-13-00875-t006:** Mediating effect test.

Relation of Path	The Point Estimate	Value of T	Bootstrap 5000 Times (95% CI)			
			Bias-Corrected		Percentile	
			Lower Limit	Upper Limit	Lower Limit	Upper Limit
IEP -> ICP -> SP -> CD -> CP	0.109	3.065	0.042	0.183	0.041	0.181
IEP -> ICP -> CP	0.291	5.2	0.182	0.402	0.185	0.404
ICP -> CS -> CP	−0.136	3.691	−0.228	−0.079	−0.221	−0.076
SP -> CD -> CP	0.151	3.183	0.06	0.247	0.057	0.246
IEP -> ICP -> CS -> CP	−0.121	3.597	−0.205	−0.069	−0.2	−0.066
CRS -> SP -> CD -> CP	0.026	2.483	0.01	0.052	0.008	0.049
CRS -> CD -> CP	0.634	9.415	0.497	0.763	0.499	0.764
ICP -> SP -> CD -> CP	0.122	3.125	0.047	0.201	0.047	0.201

Note: IEP stands for international energy price, ICP for international corn price, SP for substitute price, CD for corn demand, CP for corn price, CS for corn supply, and CRS for currency supply.

## Data Availability

The original contributions presented in the study are included in the article, further inquiries can be directed to the corresponding author.

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
