# Peer review of "Research on the Mechanism of Corn Price Formation in China Based on the PLS-SEM Model"

_foods, 2024, doi:10.3390/foods13060875_

Round 1
Reviewer 1 Report
Comments and Suggestions for Authors
-
1. Abstract: The abstract should comprehensively cover four essential elements: research background, methodology, results, and conclusions. It is crucial to present these aspects from an explanatory standpoint, addressing any gaps in content explanations.
-
2. Introduction: Some assertions in the introduction require citation to support statements, such as the significance of China's food production and its position, as well as the impact of food price fluctuations. While these points are factual, it is essential to provide sources for credibility.
-
3. Literature Review: Regarding the complexity of corn price formation mentioned in the literature review and the involvement of multiple factors, a more in-depth exploration of the mechanism behind corn price formation is recommended. This could include a detailed analysis of the interactions among different factors, changes in factor weights, and the influence of policy reforms on the formation mechanism.
-
4. Section Titles: Section 4 and 5 share the same title. Consider merging them for clarity and coherence.
-
5. Research Methods: In the research methods section, provide a detailed explanation of the advantages of using PLS-SEM for small sample sizes and non-normal data. Help readers understand why this method was chosen. When describing latent variables, briefly introduce each variable's definition and its specific significance in the study. This will aid readers in better comprehending the interrelationships of these variables and why they were chosen as key factors. Clearly explain the data source, collection methods, and address any issues related to missing data.
-
6. Conclusion: While the conclusion presents several solutions for individual issues, it is essential to strengthen the narrative on why regulating food prices is necessary and the degree of regulation needed. For instance, specify the threshold beyond which prices should be suppressed, any upper or lower limits on the price range, and the conditions necessitating policy initiatives.
-
7. Conclusion Narration: The conclusion's straightforward narration may unintentionally align with conventional observations. Acknowledge the simplicity of assumptions in the study and highlight a couple of key aspects that challenge the variation in method application, data utilization, and perspectives on problem interpretation. This will add depth and uniqueness to the conclusion, moving beyond general trends.
NaN
Author Response
please find the reply attached.

Reviewer 2 Report
Comments and Suggestions for Authors
The topic addressed in the manuscript is quite interesting in terms of research and economic methodology.
It would also be useful to include some reference to the livestock corn market, which I feel is not sufficiently considered.
The role of the money supply is not clear enough. I wonder if it cannot be removed from the model.
The implications part of the manuscript requires some in-depth study, especially to better understand the effects brought about by the abolition of the 'Temporary Corn Storage" system.
Useful enriching bibliographical references
Reviewer 3 Report
Comments and Suggestions for Authors
Line 70. Why does the author use this methodology? Is there any other paper about this subject?
The paper presents two kinds of references, by number and the ones as the reference of lines 86-87. Correct it
Line 129. The author suggests there are few papers about corn prices. There are some papers about it. Introduce them here
What’s WTO inline 203
Line 267. Indicate the source of the 84 monthly data
Table 1. Indicate the source
Indicate in another table the value of variables of Table 1 for the years 2016-23
Improve the quality of expression in lines 299-300
There is no discussion in the paper
Round 2
Reviewer 1 Report
Comments and Suggestions for Authors
From my personal perspective, I believe the article hasn't seen significant improvement, and the corresponding responses have failed to address the existing issues.
Comments on the Quality of English LanguageNaN